# Interplays of AMPK and TOR in Autophagy Regulation in Yeast

**DOI:** 10.3390/cells12040519

**Published:** 2023-02-04

**Authors:** John-Patrick Alao, Luc Legon, Aleksandra Dabrowska, Anne-Marie Tricolici, Juhi Kumar, Charalampos Rallis

**Affiliations:** School of Life Sciences, University of Essex, Wivenhoe Park, Colchester CO4 3SQ, UK

**Keywords:** fission yeast, *S. pombe*, mTOR, rapamycin, caloric restriction, lifespan, ageing

## Abstract

Cells survey their environment and need to balance growth and anabolism with stress programmes and catabolism towards maximum cellular bioenergetics economy and survival. Nutrient-responsive pathways, such as the mechanistic target of rapamycin (mTOR) interact and cross-talk, continuously, with stress-responsive hubs such as the AMP-activated protein kinase (AMPK) to regulate fundamental cellular processes such as transcription, protein translation, lipid and carbohydrate homeostasis. Especially in nutrient stresses or deprivations, cells tune their metabolism accordingly and, crucially, recycle materials through autophagy mechanisms. It has now become apparent that autophagy is pivotal in lifespan, health and cell survival as it is a gatekeeper of clearing damaged macromolecules and organelles and serving as quality assurance mechanism within cells. Autophagy is hard-wired with energy and nutrient levels as well as with damage-response, and yeasts have been instrumental in elucidating such connectivities. In this review, we briefly outline cross-talks and feedback loops that link growth and stress, mainly, in the fission yeast *Schizosaccharomyces pombe*, a favourite model in cell and molecular biology.

## 1. Introduction

Autophagy is a highly conserved homeostatic mechanism that allows cells to recycle surplus and/or damaged intracellular components. It relies on encapsulation of various cargos in a double membrane and their subsequent delivery to lysosomes (in mammalian cells) or vacuoles (in yeasts) for degradation. Upon processing, the building blocks of the broken-down cellular constituents replenish the available pool of metabolic precursors required for anabolic reactions within cells. For a long time, it was believed that autophagy is a non-selective mechanism that relies on the processing of bulk cargos. We now know that while this is still true for starvation-induced autophagy [1], selectivity has been observed in different contexts, such as cellular remodelling or disposal of damaged organelles and cellular debris [2], processes extensively reviewed elsewhere [3,4,5,6].

The term autophagy is most often used to specifically refer to macroautophagy. Nevertheless, there are also at least two other commonly distinguished subtypes: microautophagy and chaperone-mediated autophagy (CMA). The main difference between them is the location of the cargo capture site. Macroautophagy relies on the cargo being sequestered away from the lysosome/vacuole, in an autophagosome, formed *de novo*. Microautophagy and CMA both take place directly at the lysosomal membrane—the former requiring invagination of the lysosomal membrane followed by shedding of vesicles into the lumen of the organelle [7] and the latter involving the molecular chaperone complex recognition of a pentapeptide motif, KFERQ, that is present on the surface of a protein destined for degradation followed by its direct transport across the membrane [8].

Autophagy was first described by Christian de Duve in 1963 (a discovery that led to a Nobel Prize in physiology or medicine in 1974). However, it wasn’t until the 90s that the exact molecular mechanism started to be elucidated. Yoshinori Oshumi identified the first autophagy-defective (APG) yeast mutants, which are now better known as ATG genes (AuTophaGy-related), using the budding yeast *Saccharomyces cerevisiae* as a model organism [9]. Again, a discovery leading to a Nobel Prize in 2016 [10]. Around 30 different ATG genes have been described thus far in yeast model systems [11] and the highly conserved mechanistic target of rapamycin (mTOR) signalling pathway has been identified as a key regulator of autophagy, acting upstream of the ATG factors [12,13].

It is not surprising, given the role that autophagy plays in the disposal of surplus and damaged cellular material, that malfunction of this process has a role in several pathologies, such as cancer as well as immunological and neurodegenerative disorders (most notably protein conformational disorders). Compromised autophagy has been shown to be a contributing factor in Alzheimer’s disease (AD), as evidenced by accumulation of immature autophagosomes [14] and decreased expression of autophagy genes in AD brains [15]. Huntington’s disease has also been shown to display defective degradation of mitochondria and polyglutamine-expanded Huntingtin aggregates [16] (Huntingtin is a protein with no homology to other proteins, highly expressed in neurons. Nevertheless, its role is not well-understood). In the context of cancer, autophagy has a more complex role as it can either aid in the growth of tumours or act as a tumour suppressor. While disruption of beclin-1, the mammalian ortholog of yeast Atg6, has been shown to promote tumorigenesis in mice [17], and in pancreatic cancer, overactivity of the autophagosomal pathway is associated with aggressiveness and negative prognosis [18]. 

While a wealth of knowledge regarding the mechanism of autophagy has been gained from the studies performed in the budding yeast *Saccharomyces cerevisiae*, fission yeast *Schizosaccharomyces pombe* as a model system has also provided crucial and unique insights as certain aspects of the process within *S. pombe* carry more resemblance to higher eukaryotes. Fission yeast remains a powerful tool for studying conserved signalling pathways that play a key role in linking nutrient availability with cellular processes involved in the evolution of ageing [19,20,21,22,23,24] and disease [25,26] in higher organisms [27]. Almost 70% of *S. pombe* genes have human homologs [28]. In the current review, we will focus on the role played by the mTOR/AMPK axis, mostly, in yeasts and its connections to stress response and autophagy induction.

## 2. The Mechanistic Target of Rapamycin Pathway in Metabolism and Amino Acid Sensing

In this section, we present a short account of the TORC1/TORC2 complexes, their regulation through amino acids and their effects on amino acid permeases. This will help in the later discussion on circuits related to nutrient availability (glucose and nitrogen), autophagy, mTOR and the AMP-activated protein kinase (AMPK) signalling pathways.

The highly conserved mechanistic target of rapamycin (mTOR) signalling hub comprises of two distinct multiprotein complexes, TOR complex 1 (TORC1) and TOR complex 2 (TORC2) [29]. As opposed to human cells that contain a single TOR kinase, fission yeast has two Tor kinases: Tor1 and Tor2. TORC1 is composed of the essential, Tor2 that forms a complex with Mip1, Wat1/Pop3, Tco89 and Toc1. In contrast, TORC2 is composed of the Tor1 kinase that forms a complex with Ste20, Wat1/Pop3, Sin1 and Bit61 [30] (Figure 1). In response to a variety of stimuli, such as amino acids, growth factors and energy levels, TORC1 regulates metabolism and cellular growth, leading to the promotion of anabolic processes, such as protein, lipid and nucleotide synthesis, as well as the suppression of catabolic processes, such as autophagy [31] (Figure 1). Activation of TORC1 has been identified to be a two-step process (at least in mammalian cells), firstly priming, followed by activating. Prior to the stimulation of TORC1 by activating amino acids, TORC1 must be first sensitised by the following priming amino acids: Asn, Gln, Thr, Arg, Gly, Pro, Ser, Ala and Glu. Then, the activating amino acid type, such as Leu, Met, Ile and Val will interact with TORC1 and promote its activation via mTOR phosphorylation [32]. These particular experiments have been performed in human cell lines (HeLa and HEK293T cells). To reach their conclusions, researchers stimulated the cells with individual amino acids or their combinations or with an amino acid mixture that includes fifteen amino acids, and examined the status of mTOR targets, such as the S6 kinase (S6K). Experiments in *S. pombe* revealed detrimental effects of leucine withdrawal in *atg1Δ leu1-32* cells, which is more dramatic, compared to even nitrogen starvation (removal of all nitrogen sources from the media). Similar experiments in the absence of other amino acids, such as arginine or lysine, did not produce the same results, suggesting that leucine depletion itself is a potent regulator of mTOR and inducer of autophagy, and that the requirement for leucine regulates and/or contributes to autophagy, possibly through mTOR activity [33]. Mammalian TORC1 activation through amino acids is aided by recruitment of the complex on lysosomal membranes. This is achieved through recruitment of TORC1 to the RagA/B-RagC/D GTPase heterodimer. The latter is stably anchored to lysosomal membranes through the Ragulator complex [34]. While in fission yeast, a Ragulator-like complex that tethers the Gtr1-Gtr2 Rag heterodimer to the membranes of vacuoles has been identified it is not required for the vacuolar targeting of TORC1 [34]. Indeed TORC1 in *S. pombe* is thought to be resident on the vacuoles and other endosomal compartments [35]. The complex has a surprising role in attenuating TORC1 activity independently of the Tsc1-Tsc2 (tuberous sclerosis) complex. The GATOR1 complex, which functions as Gtr1 GAP, is essential for the TORC1 attenuation by the Ragulator-Rag complex, suggesting that Gtr1GDP-Gtr2 on vacuolar membranes moderates TORC1 signalling for optimal cellular response to nutrients [34].

While mTOR activity is regulated through amino acids, TORCs regulate amino acid permeases. Deletion of *tsc2* (tuberous sclerosis complex 2, negative TORC1 regulator), induces the internalization of the amino acid transporter Cat1 from the plasma membrane [36,37]. In addition, mTOR inhibition induces the transfer of several intracellular amino acid transporters, including Agp3, Isp5, Aat1 and Put4, from trans-Golgi/endosomes into the vacuoles. In these experiments, the transporters were fused with YFP [38]. In summary, constitutive Tor2 activity was shown to be critical for the retention of amino acid transporters at trans-Golgi/endosomes. Nitrogen depletion suppresses Tor2 activity through Tsc2, thereby promoting the surface expression of these transporters [38]. Beyond amino acid permease localization, mTOR seems to be regulating these proteins at the transcriptional level [39] through the GATA transcription factor Gaf1. Indeed, the loss of Tor1 decreased, and Tor2 inhibition, (through its temperature sensitive mutation), increased, mRNA expression of *isp5*, *per1*, *put4* and *SPBPB2B2.01* [39]. Gaf1 protein itself is a target of the mTOR pathway. When TORC1 is active, Gaf1 is phosphorylated and retained in the cytoplasm. Upon TORC1 inhibition (through rapamycin or torin1) Gaf1 translocates within minutes in the nucleus where it elicits its actions [40]. ChIP-seq experiments for Gaf1, combined with gene expression analyses in mTOR-active and inactive states, showed that Gaf1 mediates transcriptional effects downstream of TORC1 that are related to both PolII and PolIII and regulate tRNA expression, protein translation and metabolism and affects cellular lifespan [40]. Fission yeast TORC1 (mainly Tor2-containing) and TORC2 (mainly Tor1-containing) have opposing roles [41]. Tor1 is necessary for sexual development induced by nutrient deprivation, for amino acids uptake, and for survival in a variety of stressful conditions, such as high osmolarity, oxidative stress and low or high temperature [41,42,43]. Tor2 (TORC1) has an additional function in sexual differentiation inhibition: regardless of the dietary conditions, inactivation of Tor2 results in G1 arrest, cell differentiation, mating, and meiosis, whereas overexpression of Tor2 greatly suppresses meiosis as well as sporulation efficiency. Moreover, it has been suggested that in response to nutrient availability, Tor2 is an important regulator of the switch between cell growth and cell differentiation [44]. The two TORCs can differentially regulate amino acid permeases. For example, tuberous sclerosis complex (TSC) disruption can lead to rapamycin-sensitive phenotypes in poor nitrogen conditions. The sensitivity phenotype can be suppressed by overexpressing Isp7, a member of the family of 2-oxoglutarate-Fe(II)-dependent oxygenase genes [45]. Interestingly, *isp7* at the transcriptional level, is negatively regulated by TORC1 and positively by TORC2 [45]. Isp7 seems to have a central role in homeostasis as it regulates the expression (as well as the phosphorylation) of the ribosomal S6 protein in a fashion similar to TORC1 and opposite that of TORC2 [45].

Even though the roles of Tor2 and Tor1 kinases in fission yeast are opposing, they are also coordinated for growth, cell cycle and separase-mediated mitosis. It has been shown that Tor1 (TORC2) appears to work in tandem with mitosis by controlling CDK activation and actin localization. Under limited glucose levels, TORC2 plays an important role in the cell cycle by deciding the proper timing of the Cdc2 Tyr15 dephosphorylation and the cell size while TORC1, under those conditions, inhibits mitosis and obstructs securin-separase, both of which are required for chromosomal segregation [46]. In fission yeast, the protein phosphatase PP2A-B55 has been found to mediate the crosstalk between TORC1 and TORC2. According to genetic studies, PP2A-B55 is found downstream of TORC1, and it becomes downregulated when the activity of TORC1 lowers. When TORC1 is inactivated, TORC2-Gad8 (Gad8 being an orthologue of the human AKT serine/threonine kinases) activity is increased via a mechanism dependent on the inactivation of PP2A-B55 [47].

## 3. Autophagy and Lifespan in Yeasts

Two assays are currently used in measuring the cellular lifespan in yeasts; these are the replicative and the chronological lifespan (RLS and CLS, respectively). Replicative lifespan is the number of cell divisions a single cell, the mother cell, can undergo before reaching irreversible senescence and, then, death. RLS is a model for mitotically active cells and is primarily measured in *S. cerevisiae*, using a microfluidics device to count the number of buds a mother cell produces. *S. cerevisiae* can be utilised for RLS studies due to the asymmetric division of the cell. Whereas chronological lifespan determines the amount of time a postmitotic population remains viable and is used for both budding and fission yeasts. CLS is monitored in two separate ways, by counting the number of colony-forming units (CFUs) generated over time and by clonogenic assays [48,49]. RLS and CLS can be utilised to unravel the implications of pharmacological or genetic interventions on cellular lifespan. The role of autophagy on lifespan is one of the many implications that can be studied. Most research into lifespan extension primarily focuses on macroautophagy, and within this section, this will be our focus [50,51,52]. There is a long-standing relationship between autophagy and interventions to ageing demonstrated in several model organisms, including yeast [53]. As mentioned within the introduction, the downregulation of autophagy has been negatively associated with lifespan: in yeast studies where autophagy-related genes are knocked out, CLS is reduced, in some cases, drastically [54,55,56].

Predictably, upregulation of autophagy through stress, stress-associated genes and pharmacological interventions has shown a positive correlation with lifespan. Screens of mutant libraries of yeast identified several long-lived mutants involved in the regulation of autophagy (Slm4, Gtr1, Gtr2, Meh1 and Nvj1) in both fission and budding yeasts [48,57]. Caloric restriction (CR) is a well-known and well-published intervention to ageing, known to extend lifespan in several model organisms [58]. CLS has been extended in both *S. cerevisiae* and *S. pombe* through CR [59,60]. Autophagy has been found to mediate the extension in lifespan seen upon CR [54,55,56]. Autophagy is also intrinsically related to the lifespan-extending effects of several compounds used as interventions to ageing, such as rapamycin, spermidine, resveratrol and cucurbitacin B. Autophagy is implicated in the lifespan extending effects of these compounds through multiple pathways: the induction of autophagy through CR and rapamycin is mTOR-dependent, whilst spermidine and resveratrol induce autophagy through mTOR-independent pathways [56,61,62,63]. This emphasises the importance autophagy and cellular clearance plays within lifespan. Downstream of several pathways, autophagy is associated with longevity and induction of the process extends lifespan independent of the pathway targeted. 

Extension of lifespan through autophagy is known to act through multiple pathways and affect lifespan, indicating the convincing association this cellular process has within all aspects of cellular longevity. Gcn4 is a prime example of the implication of autophagy regulation in both ageing paradigms. Gcn4 is a transcriptional activator that allows for cellular adaptation as part of a stress response. It induces the transcription of numerous genes that mediate several biosyntheses and stress response pathways, including autophagy, downstream of the target of rapamycin (mTOR) [64]. Autophagy induction through Gcn4 not only extends lifespan as measured by CLS, but also increases replicative lifespan. Recent findings show that it is the activation of autophagy that is responsible for an increase in RLS and not a reduction in protein synthesis [65].

Experiments using pools of mutants have been proven valuable in lifespan studies and fission yeast has been no exception to this. Using DNA insertion mutagenesis, a large number of fission yeast mutants were generated [66]. This approach allowed the insertion of random barcodes to be included within each of the mutants. Cells from ~3600 barcode-tagged insertion mutants were pooled together and aged in SD medium. Barcodes were sequenced and mutants of the cyclin genes *clg1* and *pef1* were found to be long-lived. Interestingly, *pef1* is the orthologue of human *CDK5* and budding yeast *pho85* previously not reported to affect lifespan. Using a different fission yeast deletion library [67] that includes unique and defined for each mutant barcodes, aged pools of about 3400 mutants were used for barcode sequencing analysis [68]. This screen identified 341 long-lived mutants and 1246 short-lived mutants that pointed to a plethora of previously unknown ageing-associated genes, including 46 conserved but also, entirely uncharacterized genes [68]. Short-lived gene lists were enriched for autophagy, microautophagy, mitophagy and nucleophagy. Beyond the stationary phase (glucose depletion) model of CLS, fission yeast has also been used to decipher cell survival in long quiescence using the less examined (but ecologically relevant, as yeasts experience nitrogen source fluxes) nitrogen starvation model: In the absence of nitrogen and a mating partner, fission yeast cells arrest at G1/0 and reversibly enter quiescence [69,70]. Using automated spotting and re-growth tests of the aforementioned deletion library [67], a number of genes required for maintenance in G0/1 were identified, among them, genes coding for autophagy-related proteins [71]. Further analyses revealed that proteasome and autophagy coordinate for mitochondrial maintenance and survival in long quiescence [72]. Finally, barcode-based pool experiments have been conducted using nitrogen starvation. In such an experimental setup (that can last for up to 25 weeks), both known and novel long-lived mutants have been identified. Nevertheless, this experiment showed that autophagy-related genes were among the short-lived ones [73]. Taken together, the results of these studies emphasise the importance of autophagy and its role in both dividing and non-dividing cell population lifespans. 

## 4. Interplays between mTOR, AMPK and Autophagy in Fission Yeast

The AMP-activated protein kinase (AMPK) is an important regulator of cellular homeostasis and bioenergetics. AMPK surveys and balances nutrient availability and energy demands and, accordingly, coordinates anabolism and catabolism [74,75]. In mammalian cells, AMPK regulates autophagy by inhibiting TORC1 via phosphorylation of TSC2 and RAPTOR (a core component of TORC1). TORC1 phosphorylates ULK1 (an important autophagy protein homologous to yeast Atg1), preventing its activation by AMPK. TORC1 also regulates autophagy via the regulation of ULK1 stability by phosphorylating the activating molecule in Beclin-1-regulated autophagy (autophagy and Beclin-1 regulator 1, AMBRA1). Furthermore, AMPK induces autophagy by phosphorylating Beclin-1 (BECN1) in the VPS34 (vesicle-mediated vacuolar protein sorting 34) complex [76,77,78,79,80]. AMPK is highly conserved, consisting in *S. pombe* of two catalytic α subunits (Ssp2 and Ppk9), a regulatory β subunit (Amk2) and a γ subunit (Cbs2) and has been extensively detailed elsewhere [35,81]. The catalytic α subunits contain the kinase domain, while the γ subunit contains cystathionine beta synthase (CBS) domains which bind to adenosine nucleotides. The β subunit acts as a scaffold to connect the α and γ subunits [82,83]. AMP binding results in conformational changes that enable the sustained phosphorylation of Ssp2. While a clear role in regulating autophagy has been reported for the *S. cerevisiae* AMPK homologue, Snf1, its role in *S. pombe* appears to be more complex [84,85,86]. Ssp1-dependent (Ssp1 being a calcium/calmodulin-dependent CaMKK protein kinase) activation of Ssp2 results in TORC1 inhibition and accelerated entry into mitosis [87]. In recent years, several laboratories have provided deep insights into the regulation of autophagy in *S. pombe*. Crucially, these studies hint at an evolutionary diversion in the regulation of autophagy between the budding and fission yeasts [86]. When nutrients are available, the active TORC1 complex inhibits autophagy induction by phosphorylating the Atg13 subunit of the Atg1 complex. Thus, induction of autophagy requires the inhibition of TORC1 activity [88,89]. In addition to nutrient limitation, several other environmental stresses, such as phosphorus and sulphur depletion as well as endoplasmic reticulum (ER) stress also induce autophagy in *S. pombe* (reviewed in [90]). Several environmental stresses have been shown to activate AMPK/Ssp1-Ssp2 signalling. As these environmental stresses do not necessarily induce autophagy, AMPK serves to fine-tune the cellular response to specific stresses [81,87,91]. In *S. pombe*, Ssp2 modulates TORC1 activity to regulate gene expression, cell cycle progression and mating programs in response to nutrient or environmental stress [35,87,92]. Activated AMPK inhibits TORC1 activity which, in theory, relieves its inhibition of autophagy induction [33,90,93]. In *S. pombe*, nutrient deprivation and other environmental stresses induce a decline in ATP levels that lead to the activation of the Ssp1 kinase [35,81,94]. Ssp1 has been shown to be required for cell cycle progression following exposure to environmental stress and nutrient deprivation [81,87,91]. Following exposure to stress, *S. pombe* cells transiently delay cell cycle progression as a result of Sty1-mediated activation of Srk1 which in turn phosphorylates and inactivates Cdc25. Ssp1 subsequently suppresses Srk1 activity to permit cell cycle progression [95]. Ssp2 subsequently directly and indirectly inhibits TORC1, which leads to the activation of the *S. pombe* Greatwall (GW) kinase and downstream activation of the endosulfine homologue Igo1. Activated Igo1 inhibits protein phosphatase PP2A^Pab1^ leading to Cdc2 dephosphorylation and progression into mitosis independently of cell length (Reviewed in [96]). Mutants that fail to express *ssp1* or *ssp2* remain arrested in G2 under these conditions and do not advance mitosis in response to environmental stress [87]. Environmental stresses, including nitrogen deprivation, thus, result in Ssp2-dependent TORC1 inhibition and premature entry into mitosis [81,87]. As TORC1 inhibits autophagy, its inhibition by Ssp2 would be expected to play a key role in the activation of this pathway [97]. Surprisingly, Ssp2 appears not to be required for autophagy induction or survival under glucose limiting conditions [86].

When *ssp1Δ* mutants are exposed to a poor nitrogen source, they fail to accelerate advancement into mitosis. Similarly, *ppk9Δ* mutants were also defective in advancing mitosis under these conditions. Furthermore, *ssp1Δ* and *ssp2Δ* mutants have increased basal TORC1 activity which confers resistance to the ATP-competitive mTOR inhibitor torin1 [87]. Importantly, the β and γ AMPK subunits are not required for advancement into mitosis following a shift from glutamate to proline. These findings suggest that *S. pombe* Ssp2 can be activated independently of cellular AMP levels. Furthermore, these studies suggest *S. pombe* cells can fine tune their response to stresses, such as a shift to a poor nitrogen source versus nitrogen withdrawal [81,87,91]. 

Recent studies suggest that *S. pombe* cells activate autophagy as they enter stationary phase due to nutrient depletion. Glucose withdrawal, however, does not appear to induce autophagy. Rather, autophagy occurs within a specific range of glucose concentrations (0.02–0.16%) due to a shift from fermentation to respiratory metabolism [33,86]. Thus, autophagy induction depends on glucose depletion within a narrow concentration range. Curiously, Ssp2 appears not to be required for autophagy induction (as measured by CFP-Atg8 cleavage) under these conditions, nor did *ssp2* deletion influence cellular survival [86]. Given the clear role of Ssp2 in inhibiting TORC1 activity under environmental stress conditions, this finding raises the question of how autophagy is induced independently of Ssp2-TORC1 signalling. One possibility is cross-talk between the TORC1 and TORC2 complexes under conditions of low nutrient availability. Studies on the regulation of the Ght5 glucose transporter have provided important insights into the effect of nutrient availability, TORC1/2 activity and autophagy related pathways [33,46,98,99,100,101]. In stark contrast to nitrogen withdrawal, glucose deprivation does not induce Ght5 recycling in *S. pombe*. While both glucose and nitrogen limitations induce Ssp2 activation, they differentially activate TORC1 and TORC2. Hence, glucose concentrations as low as 0.1% but not nitrogen induces TORC2 (Tor1)-mediated Gad8 phosphorylation [100].

Those glucose concentrations are low enough to induce Ssp1-Ssp2 signalling [81]. Accordingly, glucose depletion inhibits the activity of TORC2 and its downstream target Gad8. When TORC2 is active, Gad8 phosphorylates the α-arrestin Aly3 to block ubiquitylation of the Ght5 glucose transporter. When Gad8 activity is inhibited, Ght5 ubiquitylation and sorting to vacuoles via the endosomal sorting complex required for transport (ESCRT) pathway is promoted [98]. Similarly, deletion of *tor1* and *gad8* results in Ght5 receptor internalization, and deletion of *aly3* blocks this effect. Interestingly, low glucose concentrations (0.06%), in contrast to nitrogen withdrawal, do not target Ght5 to vacuoles for degradation. Internalization and degradation of the Ght5 receptor following nitrogen withdrawal may, thus, replenish cellular amino acid pools under these conditions [98]. While Ssp2 appears not to be required for autophagy induction, it has been reported to regulate Ght5 expression under these conditions. Phosphorylation by Ssp2 inhibits the Scr1 transcription factor activity via nuclear export, inducing the increased expression of several genes, including *ght5* when glucose concentrations are low [85]. It should be noted however, that *ssp2* deletion does not sensitise *S. pombe* cells to low glucose concentrations [86]. Furthermore, it remains unclear how autophagy is induced in the absence of Ssp2 signalling. Cross-talk between the TORC1 and TORC2 signalling pathways has been reported previously [33,46,100]. Firstly, Ssp2 positively regulates TORC2 activity while TORC1 is partially regulated by the TORC2 complex [33,85,100]. Inhibition of TORC2 signalling also activates Gsk3 and Gsk31 which genetically interact with *ssp1*, *ssp2* and *amk2* [102,103,104,105]. Low glucose conditions attenuate the suppressive effect of Gad8 on Gsk3 and Gsk31 activity [100,102]. Increased Gsk3/ Gsk31 activity may compensate for the loss of Ssp2 expression by inhibiting TORC1 activity via the Tsc1 and Tsc2 complex [104,105]. Furthermore, TORC2 appears to suppress autophagy by enhancing TORC1 activity. As glucose deprivation inhibits TORC2 activation while enhancing Gsk3/Gsk31 activity, these conditions may promote autophagy via reduced TORC1 activation [102,105,106] (Figure 2).

## 5. Conclusions

Studies in yeast have served well in elucidating knowledge gaps in autophagy regulation, in regard to mTOR function. Nevertheless, questions remain on autophagy regulation, especially, during glucose deprivation and without the apparent contribution of the Ssp2-TORC1 axis, given that Ssp2 is required for TORC1 inhibition and subsequent autophagy induction in *S. pombe* and other organisms. Further clarifying experiments are needed to elucidate how TORC1 and TORC2 are differentially regulated through AMPK and to gain a comprehensive knowledge of TORCs activity regulation, downstream effectors and effects and recycling mechanisms involved, as they majorly contribute to cellular health and survival. 

## Figures and Tables

**Figure 1 cells-12-00519-f001:**
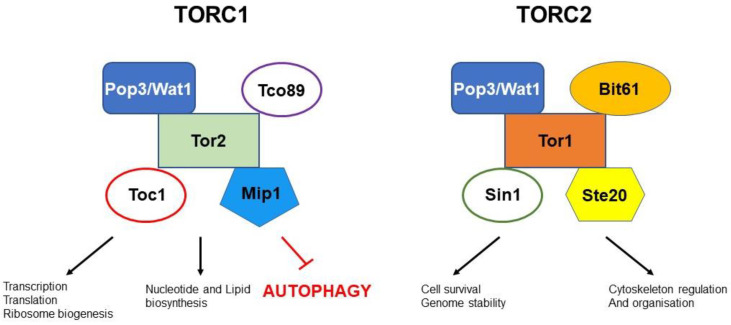
Schematic showing TORC1 and TORC2 complexes in fission yeast and the main processes they control.

**Figure 2 cells-12-00519-f002:**
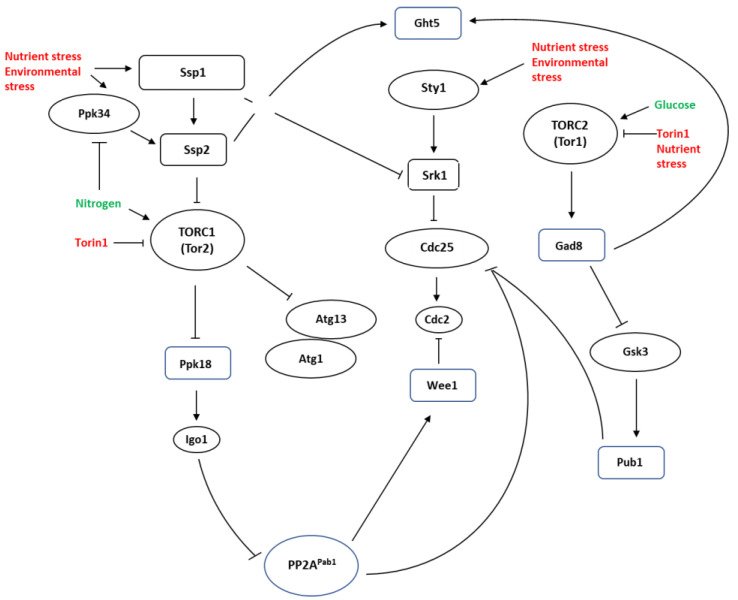
Ssp1-Ssp2 signalling integrates environmental cues and cellular ATP levels with TORC1 (mainly Tor2-containing in *S. pombe*) and TORC2 (mainly Tor1-containing in *S. pombe*) signalling. Nutrient and environmental stresses activate Ssp1 which activates Ssp2 to drive cells trough mitosis via inhibition of TORC1. Ssp1 also suppresses Srk1 stability to facilitate progression into mitosis following a transient Sty1-mediated cell cycle delay. Ssp2 inhibits TORC1 activity, resulting in the activation of the *S. pombe* Greatwall homologue Ppk18. Ppk18 activates the endosulfine Igo1 which inhibits the PP2A^Pab1^ phosphatase and drives cells into mitosis via Cdc25 activation and inhibition of Wee1. Activated Ssp2 is not required for autophagy induction but induces Ght5 hexose transporter expression when glucose levels are low. As glucose activates TORC2, Ssp2 may serve to ensure full activation of Tor1 when glucose availability is limited. Tor1 and Gad8 regulate Ght5 localisation. Gad8 also regulates amino acid transporter localisation and stability via Gsk3-mediated stabilisation of the Pub1 HECT-type E3-ligase. Nitrogen, but not glucose, withdrawal triggers Ght5 degradation. Ssp2, thus, appears to regulate Ght5 expression in response to glucose levels. It remains unclear how TORC1 is inhibited to permit autophagy induction in *ssp2Δ* mutants. Modulation of TORC2 activity under conditions of nutrient stress may influence autophagy induction. Mutants with decreased TORC2 signalling exhibit increased autophagy induction relative to wildtype cells.

## Data Availability

Not applicable.

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
