# Peer review of "Interplays of AMPK and TOR in Autophagy Regulation in Yeast"

_cells, 2023, doi:10.3390/cells12040519_

Round 1

Reviewer 1 Report

The importance of autophagy in various events has been reported in an enormous number of papers, and the current focus is on how these are regulated as an integrity of the cell or whole body. Studies have been conducted in a variety of organisms, from yeast to human, and numerous discoveries have been made. One of the major problems, however, is that findings from one organism do not necessarily apply to another.

This review summarizes findings in fission yeast on how the key actors AMPK and TOR kinases regulate autophagy in response to nutrition and other stresses, also focusing on aging. Due to differences in experimental systems, it is becoming increasingly difficult to understand research outside of one's own target organisms, and even budding yeast researchers have difficulty following the findings in fission yeast. This review article mentions similarities and differences with other organisms, and Is a good starting point for reviewing the current status and furthering one's own interests.

Author Response

The reviewer has not highlighted any particular issue with the manuscript.

Reviewer 2 Report

The manuscript of by Aloa et al focuses on the interplay between AMPK and TOR in autophagy in yeast. The manuscript is main focussing on the fission yeast. However budding yeast is often well described too. 

I think the review is very suitable for publication, but i have a few small comments.

- I think the authors should restrict themselves a bit more to fission yeast for the reasons states in the their manuscript. There is a lot of mentioning on replicative life span, but this is perhaps not relevant as this cannot be measured in fission yeast (if I am correct). Maybe keep the focus in fission yeast here as the literature of budding yeast is so extensive. 

- the authors end the manuscript like it was written for both budding and fission yeast, but it makes more sense to keep the focus on fission yeast.

- I could not find the legend of figure 1. 

- in the conclusions again I would focus on fission yeast. 

I am not expert on fission yeast, but do know the budding yeast field well. 

Author Response

Point-by-point answers to reviewer:

  1. I think the authors should restrict themselves a bit more to fission yeast for the reasons states in the their manuscript. There is a lot of mentioning on replicative life span, but this is perhaps not relevant as this cannot be measured in fission yeast (if I am correct). Maybe keep the focus in fission yeast here as the literature of budding yeast is so extensive.

We mention replicative lifespan (RLS) in order to underline the differences with chronological (CLS) and provide more context for the latter. Reports so far in fission yeast do claim that RLS is not defined in perfect (non-stress) conditions. Nevertheless, it might be that during stress fission yeast cells could also have a restricted number of mitotic divisions. 

2. the authors end the manuscript like it was written for both budding and fission yeast, but it makes more sense to keep the focus on fission yeast.

We now mention mainly fission yeast in that section.

3. I could not find the legend of figure 1. 

We apologise for this. It was a formatting error within the document and it has been mended.

4. in the conclusions again I would focus on fission yeast. 

This is now changed and we mention only fission yeast.

Reviewer 3 Report

The manuscript is a review describing the interplay between AMPK and TOR signaling in autophagy regulation in yeast.

The review summarizes current knowledge of autophagy regulation under various life conditions. The topic is interesting and such review is needed.

I have several minor comments to the manuscript:

Although the manuscript is good structured, some parts of individual sections are written in little bit confusing way. As for example in lines 95-101 it is unclear what experiments were performed and how authors of the cited study came to their conclusions. Often it is not clear whether mammalian or yeast TOR pathway is described. Could authors make it clearer?

Please add additional information about each protein/gene that is mentioned in the text and its role is not explained.

line 47 Saccharomyces cerevisiae not Schizosaccharomyces cerevisiae

line 148 figure legend to the Figure 1 is not complete, adjust the figure and text

line 176 TOC1 change for TORC1

use italics when writing S. pombe or S. cerevisiae throughout the paper

Author Response

1. Although the manuscript is good structured, some parts of individual sections are written in little bit confusing way. As for example in lines 95-101 it is unclear what experiments were performed and how authors of the cited study came to their conclusions. Often it is not clear whether mammalian or yeast TOR pathway is described. Could authors make it clearer?

The organism or system that these experiments are performed are mentioned for clarification. We have mentioned experiments that led the researchers to these conclusions.

2. Please add additional information about each protein/gene that is mentioned in the text and its role is not explained.

We have referred to some of these proteins to help the reader.

3. line 47 Saccharomyces cerevisiae not Schizosaccharomyces cerevisiae

This is now changed.

4. line 148 figure legend to the Figure 1 is not complete, adjust the figure and text

This was a formatting issue and we apologise. It has now been corrected.

5. line 176 TOC1 change for TORC1

This is now corrected.

6. use italics when writing S. pombe or S. cerevisiae throughout the paper

This is now corrected.